# TRACE Back from the Future: A Probabilistic Reasoning Approach to Controllable Language Generation

Gwen Yidou-Weng* [1]  Benjie Wang* [1]  Guy Van den Broeck [1]

## Abstract

As large language models (LMs) advance, there is an increasing need to control their outputs to align with human values (e.g., detoxification) or desired attributes (e.g., personalization, topic). However, autoregressive models focus on next-token predictions and struggle with global properties that require looking ahead. Existing solutions either post-train LMs for each new attribute—expensive and inflexible—or approximate the Expected Attribute Probability (EAP) of future sequences by sampling or training, which is slow and unreliable for rare attributes. We introduce **TRACE** (**T**ractable Probabilistic **R**easoning for **A**daptable **C**ontrollable g**E**neration)[1], a novel framework that efficiently computes EAP and adapts to new attributes through tractable *probabilistic reasoning* and lightweight *control*. TRACE distills a Hidden Markov Model (HMM) from an LM and pairs it with a small classifier to estimate attribute probabilities, enabling exact EAP computation over the HMM's predicted futures. This EAP is then used to reweigh the LM's next-token probabilities for globally compliant continuations. Empirically, TRACE achieves state-of-the-art detoxification results with only 20% decoding overhead, yields 76 low-resource personalized LMs within seconds, and seamlessly extends to composite attributes.

## 1. Introduction

As large language models (LMs) become more ubiquitous in commercial products and daily life, there is a growing need to *control* their outputs. There is a large body of re-

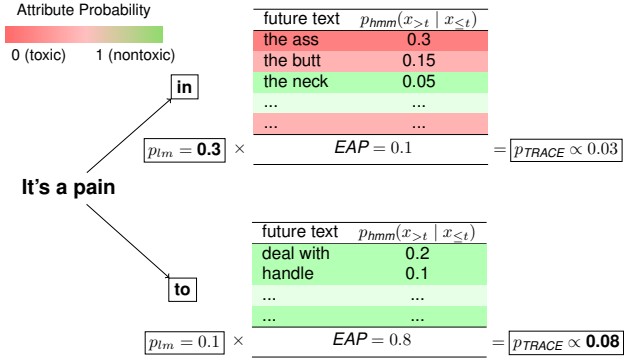

*Figure 1.* TRACE reweighs LM next token probabilities by "lookahead," computing the Expected Attribute Probability (EAP), $\sum_{x_{>t}} p(s \mid x_{\leq t}, x_{>t}) \cdot p_{hmm}(x_{>t} \mid x_{\leq t})$, using an HMM to tractably compute the expectation of a probabilistic classifier $s$.

search in LM alignment with objectives such as detoxification (Gehman et al., 2020; Xu et al., 2021), where abundant data and benchmarks exist. Meanwhile, there is growing interest involving specialized or personal attributes under low-data conditions (Adiwardana et al., 2020; Xu et al., 2021), as well as compositional attributes for creating complex or novel outputs that rarely appear during standard training (Liu et al., 2022).

Controllable text generation is challenging because most language models are autoregressive, generating each token solely from its predecessors without looking ahead—yet many attributes depend on the *entire* text. One line of solutions modifies the base model's distribution via fine-tuning or post-training (e.g., RL, RLHF) (Gururangan et al., 2020; Rafailov et al., 2023; Schulman et al., 2017), but these approaches can be highly expensive and risk degrading fluency or diversity (Kumar et al., 2022). Thus, there is a need for more lightweight solutions that can directly leverage pretrained LMs by changing the decoding strategy.

Fundamentally, we argue that decoding-based controllable generation is a *probabilistic reasoning* task: we wish to sample from the language model's distribution *conditional on some attribute*. For example, some approaches utilize sampling to find attribute-consistent generations (Yang &

---

*Equal contribution [1]Department of Computer Science, University of California, Los Angeles, USA. Correspondence to: Gwen Weng <gwenweng@ucla.edu>.

*Proceedings of the 42nd International Conference on Machine Learning*, Vancouver, Canada. PMLR 267, 2025. Copyright 2025 by the author(s).

[1]Our code is available at: https://github.com/yidouweng/trace

Klein, 2021; Tu et al., 2024; Chakraborty et al., 2024; Mudgal et al., 2024), but this can be computationally expensive. Other methods train expensive LM-based discriminators to predict satisfaction of an attribute given a partial text sequence (Krause et al., 2021; Meng et al., 2022; Liu et al., 2021), but this requires large amounts of data and training time. Both struggle with rare attributes due to limited data and high variance.

In this work, we propose a framework for controllable generation that utilizes *tractable models* to approximate the probability of satisfying an attribute given a partial sequence. This approach avoids costly sampling methods and retraining the base language model (LM) for each target attribute. First, we perform a one-time distillation of a tractable Hidden Markov Model (HMM) to approximate the base LM. Next, for each desired attribute, we train a log-linear classifier to estimate the probability of satisfying a target attribute given the entire text. During decoding, we use Bayesian conditioning to reweight next-token probabilities based on the likelihood that future sequences—predicted by the HMM—comply with these attributes, which we refer to as Expected Attribute Probability (EAP) (Figure 1). Crucially, the combination of the tractable HMM and log-linear classifier enables us to compute EAP over all of the exponentially many future sequences efficiently and exactly (over the HMM state space).

Our method, **T**ractable probabilistic **R**easoning for **A**daptable **C**onstrained g**E**neration (**TRACE**), constitutes a uniquely *lightweight* solution that enforces control with almost zero decoding-time overhead over the base LM. It further decouples generative model *training* from *control*, eliminating the need to retrain the LM or HMM for new objectives. Adapting to novel or rare attributes merely requires training a small classifier in seconds. Handling compositions of attributes is also straightforward by multiplying the EAP of each attribute during decoding.

We evaluate TRACE on three important tasks: *(1) Detoxification*: TRACE outperforms expensive RL, training- and sampling-based baselines on GPT2-large and Gemma 2B. *(2) Personalized LLMs*: TRACE adapts to 76 distinct characters in about three seconds each, outperforming prompting approaches with only a few hundred training samples, and taking only $\sim 1.2\times$ time per-token to standard decoding. *(3) Compositional Attributes*: TRACE seamlessly generates texts satisfying multiple attributes–e.g. *political* and *nontoxic*–a combination too sparse for training or sampling methods. Overall, **TRACE** is a simple, lightweight controllable generation approach that achieves state-of-the-art detoxification performance, extends to low-resource and composite text generation, and scales well to modern LLMs.

## 2. Related Work

Controllable text generation methods fall into two main categories: those that modify the language model (LM) via training, and those that steer a pre-trained model through decoding-time interventions.

### 2.1. Training Methods

One line of approaches modifies the base LM parameters, typically through **fine-tuning or reinforcement learning**, to instill desired attributes directly into the model's distribution. **DAPT** (Gururangan et al., 2020) fine-tunes the base LM on domain-specific data. **PPO** (Schulman et al., 2017) uses a reward model and policy gradients to fine-tune the LM towards desired behaviors like non-toxicity, while **DPO** (Lee et al., 2024) aligns the model using pairwise preferences without an explicit reward model. Similarly, **Quark** (Lu et al., 2022) uses an RL-like procedure conditioned on a learned reward token. A major drawback of these methods is the need for substantial data and costly retraining of the large LM for *each* new attribute or set of attributes. Furthermore, modifying the base LM risks degrading its general fluency and diversity (Kumar et al., 2022).

### 2.2. Decoding Methods

An alternative direction focuses on modifying the decoding process of a fixed, pre-trained LM to steer generation towards desired attributes, often by incorporating an estimate of the Expected Attribute Probability (EAP) of future text.

**Training Discriminators.** Since exact EAP computation requires summing over future sequences, many methods train auxiliary models to estimate attribute satisfaction from partial generations. **FUDGE** (Yang & Klein, 2021) and **NADO** (Meng et al., 2022) train discriminators to approximate EAP; **DExperts** (Liu et al., 2021) blends expert and anti-expert LMs; **GeDi** (Krause et al., 2021) uses attribute-conditioned guides with Bayes' rule; and **LiSeCo** (Cheng et al., 2024) applies a linear probe in latent space. A key challenge across these methods is that accurately estimating future attribute satisfaction from prefixes requires nontrivial lookahead, and training a separate large, auxiliary model per attribute adds significant overhead.

**Sampling.** Other methods incorporate EAP by sampling future sequences to estimate expected outcomes. Some perform limited token-level lookahead (Mudgal et al., 2024; Chakraborty et al., 2024; Tu et al., 2024), managing complexity by restricting the search space. Others employ MCMC-style sampling on entire sequences using energy-based models (e.g., **MuCoLa** (Kumar et al., 2022), **Mix and Match** (Mireshghallah et al., 2022), **COLD** (Qin et al., 2022)). Both approaches incur significant computational overhead during decoding and the resulting EAP estimates can have high variance, especially for rare attributes.

While all EAP-based control methods must approximate the intractable EAP under the base LM, the *nature* of approximation varies substantially. Discriminator-/guide-based and sampling-based approaches estimate the EAP directly during decoding, often requiring per-attribute training or incurring high variance and runtime cost. In contrast, TRACE shifts the approximation burden to a one-time HMM distillation step, where $p_{\text{hmm}} \approx p_{\text{lm}}$. Conditioned on this distilled HMM, TRACE enables an *exact and tractable* computation of EAP—$p_{\text{hmm}}(s \mid x_{<t}, x_t)$—relative to the HMM distribution (Section 4.2). This design choice—approximating the LM via a one-time HMM distillation—enables efficient, low-variance, and adaptable EAP computation during decoding, avoiding the cost of per-attribute retraining or sampling, and proves more effective in practice (Section 5.2).

## 2.3. Control via Tractable Models

Tractable probabilistic models (Choi et al., 2020) such as HMMs enable efficient computation of various quantities such as marginals and the probability of satisfying logical constraints–computations that are provably hard even to approximate on autoregressive models (Roth, 1996). Prior work has used such models to enforce *logical* or *lexical* constraints in generative modeling (Liu et al., 2024a;b). In language modeling, Ahmed et al. (2023) proposed using local tractable approximations for training autoregressive models to satisfy logical constraints. Meanwhile, Zhang et al. (2023; 2024) used HMMs to enforce logical constraints such as those given by deterministic finite automata (DFA).

While powerful for such formally specifiable constraints, this approach does not readily extend to high-level *semantic* attributes such as style, safety, or persona, which lack symbolic definitions and depend on the overall meaning of the text. Our work, TRACE, bridges this gap by employing the HMM uniquely for *semantic* control, enabling efficient *probabilistic reasoning* about semantic attributes (via EAP computation) rather than enforcing logical rules.

## 3. Preliminaries

### 3.1. Controllable Generation

We consider generating a text (sequence of tokens) $x_{1:n}$ of length $n$ from a language model (LM). In controllable generation, the goal is to generate text from the LM conditional on some attribute $s$, such as nontoxicity. We assume that the attribute can be measured by some probabilistic classifier $p(s|x_{1:n}) \in [0,1]$ representing the probability (or degree) of satisfaction of the attribute given the full text.

Let us denote the base LM distribution by $p_{lm}(x_{1:n})$. Then the joint distribution over the text $x_{1:n}$ and attribute $s$ is
$$p_{lm}(x_{1:n}, s) = p_{lm}(x_{1:n})p(s|x_{1:n}). \quad (1)$$
Our goal is then to generate from the *conditional* distri-

bution $p_{lm}(x_{1:n} \mid s)$. This conditional distribution can be decomposed autoregressively as:
$$p_{lm}(x_{1:n} \mid s) = \prod_{t=1}^{n} p_{lm}(x_t \mid x_{<t}, s). \quad (2)$$
However, sampling from the conditional next-token distribution $p_{lm}(x_t \mid x_{<t}, s)$ is generally intractable. Using Bayes' rule, this is given by:
$$p_{lm}(x_t \mid x_{<t}, s) \propto p_{lm}(x_t \mid x_{<t}) \cdot p_{lm}(s \mid x_t, x_{<t}). \quad (3)$$
The first term is simply the LM next token distribution. The second term is the probability of satisfying the attribute $s$; this requires summing over all possible continuations $x_{>t}$, which is exponential in sequence length:
$$p_{lm}(s \mid x_t, x_{<t}) = \sum_{x_{>t}} p(s|x_{\leq t}, x_{>t}) \, p_{lm}(x_{>t} \mid x_{\leq t}).$$

Since the classifier is probabilistic, we will also call this the *expected attribute probability* (EAP). The EAP is used to reweight the possible token generations $x_t$ according to how likely they are to result in a text that eventually satisfies the desired attribute. Several existing approaches effectively aim to approximate this *computationally hard* sum. For example, GeDi and DExperts train step-wise discriminators to guide the LM. Other approaches, such as Controlled Decoding and MuCoLa sample future sequences for lookahead. In this work, we aim for a tractable way to incorporate future-sequence information without expensive sampling or retraining, using Hidden Markov Models.

### 3.2. Hidden Markov Models

Hidden Markov models (HMM) specify a joint distribution over a set of latent variables $z_{1:n}$ and observed variables $x_{1:n}$, as
$$p(x_{1:n}, z_{1:n}) = p(z_1)p(x_1 \mid z_1) \prod_{t=2}^{n} p(z_t \mid z_{t-1}) \, p(x_t \mid z_t). \quad (4)$$
For language modeling, each $z_t$ takes values in $\{0, \ldots, h - 1\}$, where $h$ is the *hidden state size*, while the observed variables $x_t$ are tokens taking values in $\{0, \ldots, V - 1\}$, where $V$ is the vocabulary size. The (homogeneous) HMM has $h^2$ parameters for the transition matrix $p(z_t|z_{t-1})$, $hV$ parameters for the emission matrix $p(x_t|z_t)$, and $h$ parameters for the initial hidden state distribution $p(z_1)$. The key advantage to using HMMs for language modeling is their *tractability*; many quantities, such as the probability of a token sequence, can be inferred in linear time in the size of the HMM and sequence length. Zhang et al. (2023; 2024) distilled HMM models from large language models for the purpose of generating under logical constraints, such as the presence of a particular keyword. We will instead leverage HMMs to design algorithms for computing the expected attribute probability efficiently.

# 4. Methodology

This section details the proposed TRACE methodology. We introduce the core approximation using HMMs to guide LM generation (Section 4.1), present the algorithm enabling tractable EAP computation (Section 4.2), and describe how the attribute classifiers are fitted (Section 4.3).

## 4.1. TRACE: Guiding LM with HMM Probabilities

In order to approximate the constrained next token probability $p(x_t \mid x_{<t}, s)$ in Equation 3, we propose to approximate the expected attribute probability $p_{lm}(s \mid x_t, x_{<t})$ with the corresponding quantity under the HMM:

$$p_{\text{TRACE}}(x_t \mid x_{<t}, s) \; \propto \; p_{lm}(x_t \mid x_{<t}) \; \cdot \; p_{hmm}(s \mid x_{<t}, x_t).$$
(5)

Here, $p_{hmm}(s \mid x_{<t}, x_t)$ is the probability, under our HMM and attribute classifier, that the *entire future* continuation will satisfy the attribute. In contrast to the formulation of Zhang et al. (2023), $s$ is not a logical attribute that maps to $0$ or $1$, but instead represents a semantic attribute given by a probabilistic classifier $p(s \mid x_{1:n})$. Clearly, if the classifier $p(s \mid x_{1:n})$ is arbitrary without any structure (e.g., a neural network), then computing the expected attribute probability will be intractable as we will again need to generate all possible continuations to feed to the classifier.

Next, we describe a simple kind of classifier for which the exact computation of EAP is tractable, and then develop an efficient forward-backward style algorithm for doing so. We then describe how to learn the attribute classifier at the token level (Section 4.3), and how to improve performance further through test-time approximations.

## 4.2. Tractable Computation of EAP

The expected attribute probability (EAP) under the HMM $p_{hmm}(s \mid x_t, x_{<t})$ can be rewritten by introducing future sequence $x_{>t}$ and the hidden state $z_t$, and marginalizing over them, using the conditional independence property of HMMs: $x_{>t} \perp\!\!\!\perp x_{\leq t} \mid z_t$. We obtain that

$$p_{hmm}(s \mid x_t, x_{<t})$$
$$= \sum_{z_t} p_{hmm}(z_t \mid x_{\leq t}) \boxed{\sum_{x_{>t}} p_{hmm}(x_{>t} \mid z_t) \cdot p(s \mid x_{>t}, x_{\leq t})}.$$

We now discuss how to compute each of these terms.

**Forward Computation**  The computation of $p_{hmm}(z_t \mid x_{\leq t})$ is typically carried out using the HMM *forward algorithm*, which is based on the following recursion:

$$p_{hmm}(z_t, x_{\leq t})$$
$$= \sum_{z_{t-1}} p(x_t \mid z_t)\, p(z_t \mid z_{t-1}) \cdot p_{hmm}(z_{t-1}, x_{\leq t-1}), \quad (6)$$

with the base case $p_{hmm}(z_1, x_{\leq 1}) = p(z_1)p(x_1 \mid z_1)$. To obtain the conditional, we simply divide by the normalizing constant $p_{hmm}(x_{\leq t}) = \sum_{z_t} p_{hmm}(z_t, x_{\leq t})$ . Note that this computation is independent of the classifier.

**Backward Computation**  The quantity $p_{hmm}(x_{>t} \mid z_t)$ can be computed by exploiting the *structure* of the probability distribution in Equation 4 to rearrange the summation over future latent states $z_{>t}$:

$$p_{hmm}(x_{>t} \mid z_t) = \sum_{z_{>t}} \prod_{i>t} p(z_i \mid z_{i-1}) \cdot p(x_i \mid z_i)$$
$$= \sum_{z_{t+1}} p(z_{t+1} \mid z_t)p(x_{t+1} \mid z_{t+1}) \ldots \sum_{z_n} p(z_n \mid z_{n-1})p(x_n \mid z_n).$$

This is known as the *backward algorithm* as the evaluation of the summations is performed right to left, backward in time. Now, in order to compute the boxed term tractably, the classifier $p(s \mid x_{>t}, x_{\leq t})$ must have similar *structure* that enables integration into the backward algorithm. A sufficient condition is to restrict to *factorized classifiers* of the form $p(s \mid x_{1:n}) = \prod_i w(x_i)$, where $w(x_i)$ is a weight function that assigns a weight for each token in the vocabulary.[2]

Then, the boxed term can be expanded as

$$\boxed{\sum_{x_{>t}} p_{hmm}(x_{>t} \mid z_t) \cdot p(s \mid x_{>t}, x_{\leq t})}$$
$$= \left(\prod_{i \leq t} w(x_i)\right) \sum_{x_{>t}} p_{hmm}(x_{>t} \mid z_t) \prod_{i>t} w(x_i)$$
$$= \left(\prod_{i \leq t} w(x_i)\right) \mathbb{E}_{hmm}\left[\prod_{i>t} w(x_i) \; \Big| \; z_t\right].$$

For the expectation term $\mathbb{E}_{hmm}\left[\prod_{i>t} w(x_i) \mid z_t\right]$, we compute recursively backwards in time as follows. The base case $\mathbb{E}\left[\prod_{i>n} w(x_i) \mid z_n\right] = 1$, as there are no tokens after $x_n$. For $t < n$, the recursion is

$$\mathbb{E}_{hmm}\left[\prod_{i>t} w(x_i) \; \Big| \; z_t\right]$$
$$= \sum_{z_{t+1}} p(z_{t+1} \mid z_t) \cdot \mathbb{E}_{hmm}\left[\prod_{i>t+1} w(x_i) \; \Big| \; z_{t+1}\right]$$
$$\cdot \sum_{x_{t+1}} p(x_{t+1} \mid z_{t+1}) \cdot w(x_{t+1}). \quad (7)$$

Importantly, the values $\mathbb{E}_{hmm}\left[\prod_{i>t} w(x_i) \; \Big| \; z_t\right]$ can be pre-computed and cached in a single backward pass and reused across all generations, as they depend solely on the hidden states $z_t$ and not on the specific prefix $x_{\leq t}$.

---

[2]To ensure that the classifier outputs a value in [0, 1], we will enforce that all weights $w(x_i) \in [0, 1]$ also.

---

**Algorithm 1** TRACE: Generating $n$ Tokens

---

**Require:** HMM $p_{hmm}$, LM $p_{lm}$, Classifier $w$
**Ensure:** Generated sequence $x_{1:n}$
 1: **for** each $t$ from $n$ to 1 **do**
 2:      Pre-compute $P[t, z_t] := \mathbb{E}_{hmm}\left[\prod_{i>t} w(x_i)|z_t\right]$ by Equation (7)
 3: **end for**
 4: Initialize $s_0 \leftarrow q_0$, $x_{1:0} \leftarrow \emptyset$
 5: **for** each $t$ from 1 to $n$ **do**
 6:      Compute $p_{hmm}(z_t|x_{\leq t})$ by Equation (6)
 7:      Compute $p_{hmm}(s \mid x_{<t}, x_t)$ using $p_{hmm}(z_t|x_{\leq t})$ and $P[t, z_t]$ by Equation (8)
 8:      Sample $x_t \sim p_{hmm}(s \mid x_{<t}, x_t) \cdot p_{lm}(x_t \mid x_{<t})$ by Equation (9)
 9:      Update $x_{\leq t} \leftarrow x_{<t} \oplus x_t$
10: **end for**
11: **Return** $x_{1:n}$

---

**Integration During Generation** Algorithmically, as each new token is generated, the forward probability $p(z_t \mid x_{<t}, x_t)$ is updated based on the recursion in Equation 6. Using the precomputed backward expectations, the overall expected attribute probability is computed as

$$p_{hmm}(s \mid x_t, x_{<t})$$
$$= \left(\prod_{i \leq t} w(x_i)\right) \sum_{z_t} p_{hmm}(z_t|x_{\leq t}) \cdot \mathbb{E}_{hmm}\left[\prod_{i>t} w(x_i) \mid z_t\right]. \tag{8}$$

Plugging this into Equation 5, we can then compute the constrained next-token distribution:

$$p_{\text{TRACE}}(x_t \mid x_{<t}, s) \propto p_{lm}(x_t \mid x_{<t}) \cdot p_{hmm}(s \mid x_t, x_{<t}) \tag{9}$$

Algorithm 1 summarizes our approach. The precomputation requires $O\left(n(hV + h^2)\right)$ time (backward HMM pass) and requires $O(nh)$ cache memory, corresponding to each hidden state value at each time. During generation, the computation of the expected attribute probability $p_{hmm}(s \mid x_t, x_{<t})$ at each time point takes $O(h^2 + hV)$. In practice, we find this takes negligible time relative to computing the language model's next token probability (see Table 4).

### 4.3. Attribute Classifier Fitting

As discussed, one class of classifiers that make EAP computation tractable is a product over tokens, that is, linear in log space. In practice, attributes are not fully factorizable (to varying extents); nonetheless, we show that even such a simple classifier, when combined with a HMM, captures EAP sufficiently accurately to outperform baselines using neural classifiers (e.g., GeDi, FUDGE) (See 5.2).

We assume access to an oracle $p_{\text{oracle}}$ that scores the probability of a sequence $x_{1:n}$ satisfying a target attribute $s$. This oracle may be a trained classifier (e.g., from human annotations) or an external API scoring attribute satisfaction.

**Fitting Rare attributes via Log-MSE** Our goal is to model attributes that are *rare* in natural text, such as toxicity or political content. To effectively identify these cases, the learning objective must distinguish texts that exhibit the rare attribute from the more common, neutral ones. Standard objectives like cross-entropy treat all misclassifications symmetrically, which is suboptimal when one class is rare.

We therefore use an asymmetric loss that more heavily penalizes misclassifying the rare examples. In particular, we use mean squared error loss in log-space (*log-MSE*), which amplifies error penalties in the low-probability region. To align with this objective, we define each attribute by its *absence* (e.g., modeling "nontoxicity" rather than "toxicity").

Specifically, for our factorized (log-linear) classifier, the log-probability of the attribute $s$ given the text $x_{1:n}$ is a sum of log-weights, $\log p(s \mid x_{1:n}) = \sum_{i=1}^n \log w(x_i)$. Then, given oracle scores $p_{\text{oracle}}$ (potentially transformed, as discussed next), the loss for a text $x$ is given by

$$\left\| \log p_{\text{oracle}}(x) - \sum_{i=1}^n \log w(x_i) \right\|^2 \tag{10}$$

For example, this loss penalizes a prediction of $0.5$ much more heavily if the true oracle score is $p_{\text{oracle}} = 0.1$ (toxic) than if it is $0.9$ (nontoxic), whereas cross-entropy is indifferent to these outcomes.

**Probability Transformation for Better Control** In practice, it can be beneficial to transform raw oracle probability scores ($p_{\text{oracle}}$) for a clearer separation between desired and undesired outputs, and enforce stricter attribute satisfaction. For instance, a 20% non-toxicity score from an oracle can be transformed to a lower value to encourage safer generation.

To do this, we propose to apply an affine transformation to the oracle scores in logit space using a (non-negative) scale ($b$) and shift ($c$):

$$p'_{\text{oracle}} = \sigma\left(b \cdot \ln\left(\frac{p_{\text{oracle}}}{1 - p_{\text{oracle}}}\right) + c\right),$$

where $\sigma(z)$ is the sigmoid function. This transformation reshapes the target distribution; specifically, increasing the scale $b$ pushes intermediate probabilities towards the extremes of 0 and 1. This creates a more bimodal distribution, which helps to clearly distinguish between attribute-compliant and non-compliant texts.

This probability transformation can also be applied at decoding time to the EAP, $p_{\text{hmm}}(s \mid x_t, x_{<t})$, before it influences token generation (Eq. 9). This can be used to sharpen the

*Table 1.* Detoxification on RealToxicityPrompts (Gehman et al., 2020). Results are shown for GPT-2 (10k prompts) and Gemma-2B (1k prompts). **TRACE** applies training- and decoding-time transformation (see Section 4.3). Struck-through fluency indicates unnatural repetition (Holtzman et al., 2020). Baselines: (1) Gururangan et al. (2020); (2) Krause et al. (2021); (3) Yang & Klein (2021); (4) Liu et al. (2021); (5) Dathathri et al. (2020); (6) Kumar et al. (2022); (7) Schulman et al. (2017); (8) Lu et al. (2022); (9) Lee et al. (2024) (our implementation).

| Model | Toxicity (↓) | | Diversity (↑) | | PPL (↓) | Approach Type |
| --- | --- | --- | --- | --- | --- | --- |
| | avg. max. | prob. | dist-2 | dist-3 | | |
| GPT-2 Large Results | | | | | | |
| GPT2 | 0.385 | 0.254 | 0.87 | **0.86** | **25.57** | Baseline |
| DAPT[1] | 0.428 | 0.360 | 0.84 | 0.84 | 31.21 | Finetuning |
| GeDi[2] | 0.363 | 0.217 | 0.84 | 0.83 | 60.03 | Decoding (Trained Guide) |
| FUDGE[3] | 0.302 | 0.371 | 0.78 | 0.82 | ~~12.97~~* | Decoding (Trained Guide) |
| DExperts[4] | 0.314 | 0.128 | 0.84 | 0.84 | 32.41 | Decoding (Trained Guide) |
| PPLM[5] | 0.520 | 0.518 | 0.86 | 0.86 | 32.58 | Decoding (Logit Control) |
| MuCoLa[6] | 0.308 | 0.088 | 0.82 | 0.83 | 29.92 | Decoding (Sampling) |
| PPO[7] | 0.218 | 0.044 | 0.80 | 0.84 | ~~14.27~~* | RL |
| Quark[8] | 0.196 | 0.035 | 0.80 | 0.84 | ~~12.47~~* | RL |
| DPO[9] | 0.180 | 0.026 | 0.76 | 0.78 | ~~21.59~~* | RL |
| **TRACE** | **0.163** | **0.016** | 0.85 | 0.85 | 29.83 | Decoding (HMM Reasoning) |
| Gemma-2B Results | | | | | | |
| Gemma-2B | 0.359 | 0.23 | 0.86 | 0.85 | **15.75** | Baseline |
| DPO[9] | 0.222 | 0.06 | 0.74 | 0.77 | ~~14.39~~* | RL |
| **TRACE** | **0.189** | **0.02** | **0.86** | **0.85** | 17.68 | Decoding (HMM Reasoning) |

separation between next tokens $x_t$ based on their EAP, thus ensuring stricter control of the attribute.

In Section 5.2, we empirically justify this intuition and carefully study the effect of these *training-time* and *decoding-time* probability transformations.

# 5. Experiments

We evaluate TRACE on a range of challenging controllable generation tasks: detoxification, low-resource role-playing, and compositional control involving political and non-toxic text attributes. This section details the experimental setup and presents the main results for each task, along with an analysis of TRACE's efficiency and properties.

## 5.1. Experimental Setup

**Evaluation Metrics.** Metrics vary by task. For the primary **detoxification** task (Section 5.2), we follow the setup in Liu et al. (2021) using the RealToxicityPrompts dataset (Gehman et al., 2020). We evaluate **Toxicity** (Perspective API avg. max. toxicity & prob. of any toxic generation ($> 0.5$) over 25 samples; ↓ lower is better), **Perplexity** (as an automatic measure of fluency, calculated using GPT2-XL; ↓ lower is better), and **Diversity** (Distinct 2-grams, 3-grams; ↑ higher is better). We also include supplementary AI evaluations using GPT4o-mini. For **role-playing** (Section 5.3), we measure *role quality* via classifier probability. For **topic control** (Section 5.5), we measure *political relevance* using the zero-shot classifier from Laurer et al.

(2023). Full details on metrics are provided in Appendix F.

**Baselines.** TRACE is compared against representative baselines including fine-tuning (DAPT), RL (PPO, Quark, DPO), and various decoding methods (PPLM, GeDi, FUDGE, DExperts, MuCoLa). Results are sourced primarily from prior work (Liu et al., 2021; Lu et al., 2022; Kumar et al., 2022) or run by us (DPO) using the same setup as in Liu et al. (2021). Full details are in Appendix E.

**Implementation Details.** The HMMs used by TRACE were distilled[3] from base LMs (GPT2-Large, Gemma-2B) following Zhang et al. (2023); Liu et al. (2023); details on the different HMM configurations, including hidden state sizes and training parameters, are in Appendix C. Once distilled, the HMM model is fixed and is reused without further training for each attribute. Attribute classifiers (e.g. nontoxicity) were trained according to the method in Section 4.3; datasets, oracles, fitting procedures, and transformation parameters $(b, c)$ are detailed in Appendix D.

## 5.2. Detoxification

We evaluate TRACE on detoxification using RealToxicityPrompts (Gehman et al., 2020), comparing performance against fine-tuning (DAPT), RL (PPO, Quark, DPO), and decoding methods (GeDi, FUDGE, DExperts, MuCoLa).

---

[3] The distillation process involves compiling a HMM to an equivalent probabilistic circuit (Choi et al., 2020) on the GPU, which is trained on LM samples using a mini-batch variant of the expectation-maximization (EM) algorithm (Dempster et al., 1977).

**TRACE Outperforms Existing Methods** Table 1 shows TRACE significantly reduces toxicity compared to baselines, and maintains high diversity, with minor reductions from GPT-2's baseline scores. RL methods (e.g., PPO, Quark, DPO), despite lowering toxicity, sharply reduce diversity and perplexity, indicative of repetitive or unnatural text generation due to mode collapse. Appendix Tables 7 and 8 further support these observations: TRACE achieves higher conditional entropy than DPO, and receives comparable or better ratings from GPT-4 on nontoxicity and diversity.

**Factorized Classifier is Effective Despite Non-Factorizable Attributes.** Attributes like toxicity are not fully factorizable, as evidenced by the performance gap between neural and our factorized classifiers (Appendix D). Nonetheless, TRACE outperforms methods like GeDi and FUDGE that use more expressive neural classifiers but rely on approximate EAP estimation. This highlights a key insight: exact EAP computation over an HMM, even with a simple factorized classifier, can rival or surpass more complex classifiers when the inference framework is less precise.

**Scaling TRACE to Larger Language Models.** TRACE scales effectively to larger language models such as Gemma-2B. As shown in Table 1 and supported by GPT-4 evaluations (Appendix Table 8), TRACE consistently outperforms DPO—a reinforcement learning-based method—in detoxification performance, while maintaining strong fluency and diversity. These results demonstrate TRACE's broad applicability and robustness across model scales.

*Table 2.* Ablation study of TRACE variants on GPT-2 Large Rows: no transformation, training-time transformation only, or both training- and decoding-time transformation.

| TRACE Variant | Toxicity (↓) | | Diversity (↑) | | PPL (↓) |
|---|---|---|---|---|---|
| | avg. max. | prob. | dist-2 | dist-3 | |
| No Transformation | 0.353 | 0.196 | **0.87** | **0.86** | **25.44** |
| Training TF | 0.187 | 0.026 | **0.87** | 0.85 | 27.51 |
| Train + Dec TF | **0.163** | **0.016** | 0.85 | 0.85 | 29.83 |

**Complementary Roles of Training and Decoding Transformations** The probability transformation offers complementary benefits when applied at different stages of TRACE, as shown by the ablation study in Table 2. Applying the transformation at training time alone ("Training TF") provides an initial reduction in toxicity over the baseline TRACE model with no transformation. The most significant gain comes from applying the transformation at both stages ("Train + Dec TF"); this further reduces average maximum toxicity from 0.187 to 0.163, albeit with a corresponding increase in perplexity. Figure 2 visually explains the complementary mechanisms driving these gains.

**Top:** At training time, the transformation creates a stricter

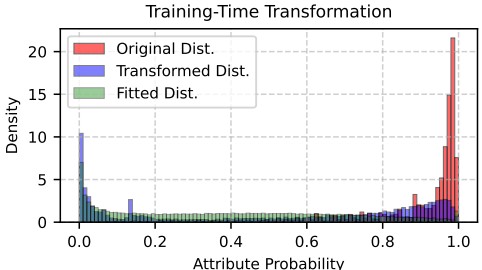

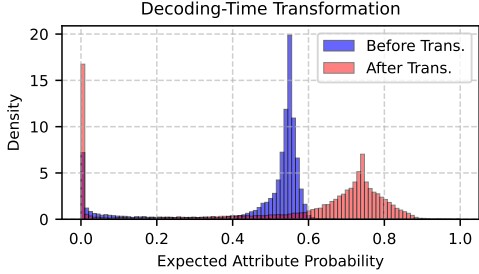

*Figure 2.* **Complementary Effects of Probability Transformation. Top:** At training time, the transformation concentrates oracle probabilities for undesirable attributes towards 0 to improve classifier *learning*. **Bottom:** At decoding time, it reshapes the unimodal EAP into a bimodal one to enforce stricter generation *control*.

learning target by reshaping the skewed Original Dist." into a bimodal "Transformed Dist.", critically pushing scores for toxic content towards 0. The log-MSE objective's focus on this low-probability region helps the classifier learn stronger negative weights for toxic tokens, resulting in a "Fitted Dist." that successfully captures this new peak near 0.

**Bottom:** At decoding time, the same transformation is applied directly to the computed Expected Attribute Probability (EAP). It converts a diffuse, unimodal EAP distribution ("Before Trans.") into a sharp, bimodal one ("After Trans."), creating a clearer separation between attribute-compliant and non-compliant continuations.

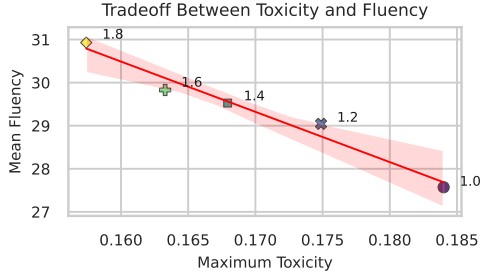

*Figure 3.* **Tuning the Fluency-Detoxification Trade-off.** The decoding-time transformation multiplier acts as a control knob; as it increases, detoxification improves while fluency decreases.

*Table 3.* Personalized LM Responses to the Question: "How do you define freedom?"

| Question | How do you define freedom? | | |
|---|---|---|---|
| **Direct Prompt** | *It's a topic we deal with very frequently and yet it's a very subjective question with no easy answer. On its surface, the idea of freedom...* | | |
| **Roles** | **Abraham Lincoln** | **Thor** | **Twilight Sparkle** |
| **Prompted with Role-Specific Instructions** | *Freedom is a fundamental human right that is necessary for individuals to live full and meaningful lives. It is the ability to make choices...* | *Freedom is more than just a freedom of speech or movement; it's a feeling of being in control of one's own destiny...* | *Freedom is a powerful concept, and it's something that I've thought a lot about throughout my adventures in Equestria...* |
| **TRACE** | **Decisive freedom, subjective freedom, freedom of speech**? There are different definitions for what freedom of speech means, but most of them are centered around the idea that it... | **Loki**, the son of **Odin**, reminds of what it is to have freedom in the eyes of a **God**. The **mighty Norse god** of mischief is known for being... | **Friendship, connection, love**... freedom is endless and depends on the person you ask. Some people want their **friends** to be real while others value their ideas and thoughts |

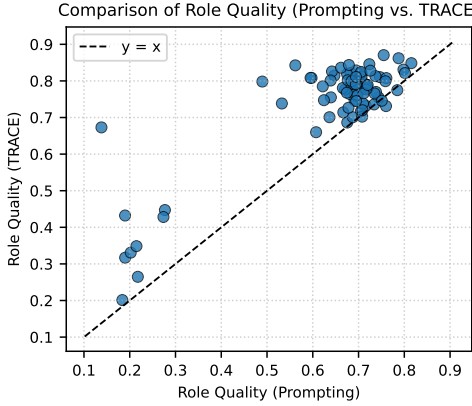

*Figure 4.* TRACE outperforms prompt engineering in role quality.

This synergy—where one transformation aids *learning* and the other enforces *control*—accounts for their strong combined performance. As a final benefit, the decoding transformation's scale parameter ($b$) offers an intuitive knob for tuning this control, allowing users to smoothly trade toxicity for fluency post-hoc (Figure 3).

**Impact of HMM Quality** Finally, the effectiveness of control is sensitive to the quality of the HMM itself. As shown in Appendix Fig. 5, as the distilled HMM's log-likelihood improves with more training, we observe a corresponding decrease in generation toxicity. This highlights that better HMM distillation directly enhances TRACE's performance.

### 5.3. Role-playing 76 Characters with Limited Data

**Lightweight and Low-Resource Adaptation.** A key advantage of TRACE is its rapid, low-resource adaptation; new attributes are integrated by training only a lightweight log-linear classifier. Leveraging this, we personalized both GPT2-large and Gemma-2B for 76 distinct characters from

the RoleBench dataset (Wang et al., 2024). Each character's classifier was trained on its corresponding RoleBench training split of $\sim 300$ question-answer pairs.

Table 3 qualitatively compares Gemma-2B responses for three characters across three settings: the base model, a few-shot prompting baseline, and TRACE. The prompting baseline uses a role instruction plus 10 question-answer examples from the RoleBench training set (details in Appendix E). Qualitatively, TRACE-guided answers show more distinct, character-reflective content and tone than the baselines.

Quantitatively, on GPT2-large, TRACE achieves superior role quality over a standard prompting baseline for most of the 76 characters (Fig. 4) . This baseline uses a role-specific but exemplar-free instruction (details in Appendix E), ensuring a direct comparison of instruction-following with minimal input overhead, unlike few-shot prompting which has higher decoding costs (see Section 5.4, Table 4).

### 5.4. Training and Inference Time Analysis

*Table 4.* Training and inference time comparison. TRACE achieves minimal training overhead and near-baseline inference cost.

| Method | Train Time | Method | Inf. Ratio |
|---|---|---|---|
| | | Baseline | 1.0 |
| Mix and Match | 2 hr | Prompting | $\sim 3.0$ |
| DExperts | 3 min–16 hr | GeDi / DExperts | 2.0–3.0 |
| DAPT | 16 hr | Mix and Match | 7.5 |
| GeDi | 5 hr | MuCoLa | 15–20 |
| **TRACE** | **10 s** | PPLM | 40.0 |
| | | **TRACE** | **1.2** |

TRACE is designed for rapid adaptation and efficient inference, requiring only a one-time HMM distillation from a base LM, independent of any specific control attribute.

**Training Time.** Once the HMM is trained, adapting to a

*Table 5.* Composition results (Political+Nontoxic) on RealToxicityPrompts with GPT2-Large. Metrics: Tox (avg. max./prob.$>0.5$; $\downarrow$), Mean Pol $\uparrow$, PPL $\downarrow$, Diversity (Dist-2/3; $\uparrow$). "+ Dec. TF" applies decoding TF (Sec 4.3) only to the Political EAP.

| Models | Max Tox ($\downarrow$) | Any Tox $> 0.5$ ($\downarrow$) | Mean Pol ($\uparrow$) | PPL ($\downarrow$) | Dist-2 ($\uparrow$) | Dist-3 ($\uparrow$) |
|---|---|---|---|---|---|---|
| GPT2-L (Base) | 0.386 | 0.257 | 0.169 | 25.74 | 0.87 | 0.86 |
| TRACE (Detox only) | 0.186 | 0.026 | 0.168 | 27.33 | 0.87 | 0.85 |
| TRACE (Pol only) | 0.379 | 0.244 | 0.333 | 29.32 | 0.87 | 0.86 |
| TRACE (Detox + Pol) | 0.190 | 0.027 | 0.344 | 29.71 | 0.87 | 0.86 |

new attribute only requires fitting a lightweight classifier. This process takes up to 10 seconds, in sharp contrast to baselines like DAPT, DExperts, GeDi, and Mix and Match, which requires anywhere from minutes to 16 hours of training on various GPUs. This rapid adaptation makes TRACE highly suitable for dynamic or low-resource scenarios.

**Inference Time.** TRACE maintains efficiency during generation. While alternative approaches that rely on in-context learning, discriminators, or iterative updates incur substantial decoding costs—with inference ratios ranging from 7.5x to 40x the baseline—TRACE introduces only minor overhead. This efficiency stems from its design; the HMM's Expected Attribute Probability (EAP) calculation uses a precomputed backward pass and an efficient forward update per token. Consequently, the total inference time is only about 1.2x that of the baseline language model, as the process remains dominated by the base model's computation.

### 5.5. Composition: Political and Nontoxic Texts

A key benefit of TRACE is its ability to compose multiple attributes without retraining. Conditioning on the conjunction of two attributes $(s_1, s_2)$ is achieved by multiplying their probabilities during decoding, based on the modeling assumption of attribute independence given the text ($x$): $p(s_1 \text{ and } s_2|x) = p(s_1|x)p(s_2|x)$. For TRACE's factorized classifiers, this simplifies to creating a new composite classifier by multiplying the token weights ($w$) of the individual classifiers: $w' = w^1 \cdot w^2$. To demonstrate, we task the model with generating text that is simultaneously **political** and **nontoxic**. This combination is rare, making joint training challenging for other methods, but TRACE remains effective as it only needs the individually trained classifiers.

The results in Table 5 demonstrate effective composition. Controlling for a single attribute primarily affects only that dimension: `TRACE (Detox only)` reduces Max Toxicity from 0.386 to 0.186 while leaving the Mean Political score nearly unchanged (0.169 vs 0.168). Conversely, `TRACE (Pol only)` increases the Mean Political score from 0.169 to 0.333 with almost no change in toxicity. The compositional `TRACE (Detox + Pol)` approach successfully achieves both goals simultaneously, reaching a toxicity level (0.190) and a political score (0.344) comparable to their respective single-attribute control settings.

## 6. Conclusion

We introduced **TRACE**, a lightweight framework for controllable text generation that uses tractable probabilistic inference. Its core strengths are efficiency, adaptability, and strong performance, all achieved without modifying or finetuning the base LM. TRACE distills a Hidden Markov Model (HMM) from a base LM and combines it with simple, efficiently trained classifiers to tractably compute the Expected Attribute Probability (EAP), which guides generation towards desired attributes.

Empirically, TRACE achieves state-of-the-art detoxification with only a $\sim$20% decoding overhead. It also excels in low-resource scenarios, adapting to personalize generation for 76 distinct characters in seconds, and successfully composes multiple attributes, such as generating text that is simultaneously political and non-toxic.

Despite TRACE's already strong empirical performance, its capabilities can be further boosted by improving the expressivity of both the HMM and the attribute classifier. Our results show that higher-quality HMMs yield better results, suggesting that improved distillation techniques are a promising direction. Additionally, while our simple factorized classifiers are effective even for non-factorizable attributes, extending TRACE to incorporate more expressive models—such as tractable probabilistic circuits (Khosravi et al., 2019; Choi et al., 2020)—that remain compatible with efficient EAP computation could enable stronger control over complex attributes like long-range coherence.

## Acknowledgements

This work was funded in part by the DARPA ANSR, CODORD, and SAFRON programs under awards FA8750-23-2-0004, HR00112590089, and HR00112530141, NSF grant IIS1943641, and gifts from Adobe Research, Cisco Research, and Amazon. Approved for public release; distribution is unlimited.

## Impact Statement

This paper presents work whose goal is to advance the field of Machine Learning. There are many potential societal consequences of our work, none which we feel must be specifically highlighted here.

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

## A. Factorizability of Attributes

To illustrate the extent to which the attributes evaluated in this work deviate from the factorized assumption made by TRACE's classifier (Section 4.3), Table 6 compares the best achievable fit of the factorized classifier against a more expressive neural classifier. Performance is measured via Cross-Entropy (CE) loss relative to oracle scores (lower indicates better fit).

*Table 6.* Attribute Factorizability: Gap Between Factorized and Neural Classifiers' Fit to Oracle Scores.

| Attribute | Factorized Classifier CE Loss | Neural Classifier CE Loss |
|---|---|---|
| Toxicity | 0.386 | 0.007 |
| Politics | 0.0064 | 0.0003 |

The results indicate that both Toxicity and Politics exhibit non-factorizable characteristics to varying degrees, as expected for complex semantic attributes. Nonetheless, as discussed in Section 5.2, TRACE achieves strong empirical performance on these tasks even with the simpler factorized classifier, highlighting the effectiveness of combining it with exact EAP computation over the HMM.

## B. Additional Detoxification Metrics

To further support the findings in Section 5.2, we provide two supplementary evaluations.

**Conditional Entropy.** Table 7 reports the conditional entropy of continuations given a prompt under each model. Lower entropy indicates less lexical and structural diversity, often symptomatic of repetitive or degenerate outputs. TRACE achieves entropy comparable to the base LM and much higher than DPO, reinforcing that RL methods tend to reduce generation diversity.

*Table 7.* Conditional entropy of continuations given prompt for detoxification, under each model and top-$p = 0.9$ sampling.

| Method | Entropy ($\uparrow$) |
|---|---|
| GPT2-large | 52.06 |
| DPO | 39.52 |
| TRACE | 52.54 |

**GPT-4 LM-as-a-Judge Evaluation.** Table 8 presents human-aligned evaluations from GPT-4, comparing continuations across nontoxicity, fluency, and diversity. TRACE matches or exceeds DPO on nontoxicity and diversity, while maintaining comparable fluency to both Gemma2B and DPO, affirming that TRACE achieves strong controllability without harming output quality.

*Table 8.* GPT4 LM-as-a-judge Evaluations.

| Method | Nontoxicity ($\uparrow$) | Fluency ($\uparrow$) | Diversity ($\uparrow$) |
|---|---|---|---|
| Gemma2B | 4.39 | 3.76 | 2.93 |
| DPO | 4.65 | 3.94 | 2.86 |
| TRACE | 4.69 | 3.72 | 2.94 |

**Impact of HMM Quality** The quality of the distilled Hidden Markov Model (HMM) has a direct impact on the effectiveness of TRACE for controllable generation. To illustrate this, Figure 5 plots the HMM's log-likelihood on a validation set against the average maximum toxicity of the generations it produces at different points during its training. The results clearly show that as the HMM becomes a better probabilistic model of the language (higher log-likelihood), its ability to guide the generation towards non-toxic outputs improves (lower toxicity). This suggests that further advancements in HMM distillation techniques will likely yield even better performance from TRACE.

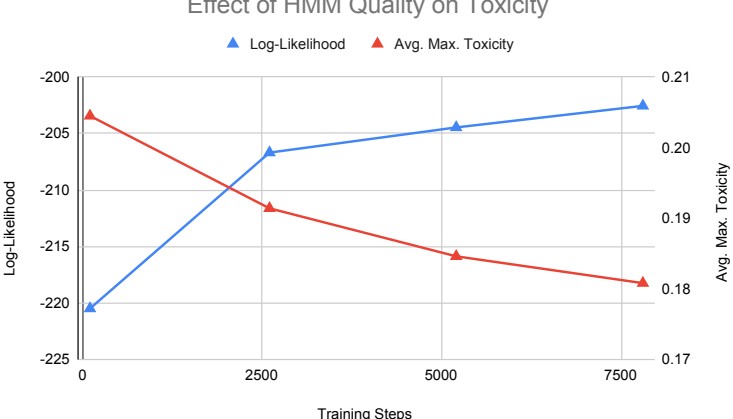

*Figure 5.* **Effect of HMM Quality on Detoxification.** As the HMM is trained for more steps, its fit to the data improves (increasing Log-Likelihood, blue), which corresponds to a decrease in the average maximum toxicity of generated text (red). This shows that higher-quality HMMs lead to more effective control.

## C. HMM Implementation Details

We employ standard Hidden Markov Models (HMMs) for all TRACE experiments, distilled from the base language models (GPT2-large, Gemma-2B) following the approach described in Zhang et al. (2023); Liu et al. (2023).

**Parameters and Training Data.** Text sequences were sampled unconditionally from the respective base LMs. We used two HMM configurations depending on the task. For the primary detoxification and composition tasks, the HMM hidden state size was set to $h = 4096$. For the low-resource personalization experiments (Section 5.3), a smaller HMM with $h = 256$ was used to demonstrate TRACE's effectiveness even with a more compact model.

Training sample sizes for the larger ($h = 4096$) HMM were:

**GPT2-large HMM:** 10 million samples.

**Gemma-2B HMM (Standard):** 10 million samples.

**Training Details.** We initialize the parameters of the HMM using the latent variable distillation technique (Liu et al., 2023). For training of the HMM, we employed a mini-batch variant of expectation maximization (EM) (Sato, 1999; Peharz et al., 2020) which interpolates between the old and new parameters (based on a mini-batch) using a step size $\alpha$. We use a mini-batch size of 4096 and train for 50 epochs, and anneal the step-size according to a linear decay schedule from 1.0 to 0.0 following Zhang et al. (2025).

**Training Time.** For the standard 10M-sample, 4096-state HMM used with GPT2-large, the process involved approximately 18 hours for text sampling plus 2 hours for HMM training itself on a single NVIDIA RTX A6000 GPU.

## D. Attribute Classifier Implementation Details

Attribute classifiers were trained following the methodology in Section 4.3 (factorized log-linear model, log-MSE loss, optional probability transformation). Specifics for each attribute were:

**Non-Toxicity Classifier.** Training data was generated by prompting GPT-2 Large with training split prompts from RealToxicityPrompts (Gehman et al., 2020). Generated continuations were scored using the Perspective API, providing the target non-toxicity probabilities $p$. The log-linear weights $l(x_i)$ were fitted using log-MSE loss against these scores after applying the logit transformation (Section 4.3) with scale $b = 10$ and shift $c = 3$.

**Role Classifier.** For the 76 characters used in Section 5.3, classifiers were trained on the RoleBench dataset (Wang et al., 2024). Specifically, the training split provides approx. 300 question-answer pairs per character, which were used as positive examples for that character's classifier. Fitting used log-MSE loss.

**Non-Politicalness Classifier.** As RealToxicityPrompts rarely elicit political content, training data was sourced from the News Category dataset (Misra, 2022). Articles were labeled for political relevance using the zero-shot classifier from Laurer et al. (2023), providing target probabilities $p$. We modeled *non-politicalness* (1-p) and fitted the log-linear weights using log-MSE loss against these scores after applying the logit transformation (Section 4.3) with scale $b = 1$ and shift $c = -10$.

# E. Baseline Details

**Sourced Baseline Results.** Performance results for prior work baselines presented in Table 1 (excluding our implementations) were sourced to ensure comparability under the DExperts setup (Liu et al., 2021), which uses 10k RealToxicityPrompts test prompts and top-$p = 0.9$ sampling (25 generations/prompt):

- DExperts, PPLM, DAPT, GeDi results from Liu et al. (2021).

- MuCoLa, FUDGE results from Kumar et al. (2022).

- PPO, Quark results from Lu et al. (2022).

Training times and inference time ratios (Table 4) for baselines were sourced from:

- *Inference Ratios:* DExperts, DAPT, GeDi, PPLM from Liu et al. (2021); Mix and Match, FUDGE from Mireshghallah et al. (2022); MuCoLa from Kumar et al. (2022).

- *Training Times:* DExperts, DAPT from Liu et al. (2021); Mix and Match from Mireshghallah et al. (2022); GeDi from official repository (`https://github.com/salesforce/GeDi`).

**Implemented Baselines.** We implemented the following for direct comparison:

Direct Preference Optimization (DPO): The original DPO paper (Lee et al., 2024) used different settings; we adapted the official implementation for GPT2-large/Gemma-2B and evaluated under the DExperts setup for comparability.

Prompting Baselines for Role-Playing: For Section 5.3, we used two strategies. **Role-Specific Instruction + 10 QA** (for Gemma-2B qualitative eval, Table 3) prepended a role instruction and 10 QA examples from RoleBench (Wang et al., 2024) to the query (see example structure below). **Role-Specific Instruction Only** (for GPT2-large quantitative eval, Figure 4) used only the instruction text.

---

**EXAMPLE FEW-SHOT ROLE-PLAYING PROMPT (ABRAHAM LINCOLN)**

You are an advanced role-playing assistant trained to embody characters with accuracy and authenticity. In this instance, you will assume the persona of Abraham Lincoln. Example Interactions:
**1. Question:** Abraham Lincoln, how did your personal tragedies and political opposition affect your commitment to justice and equality during your presidency?
**Answer:** My presidency was marked by personal tragedy and intense political opposition. The loss of my son, Willie, was particularly devastating, and it weighed heavily on my mind during my time in office. Additionally, I faced intense opposition from those who were resistant to change and the abolition of slavery. However, I remained steadfast in my commitment to justice and equality, and I firmly believed that these principles were worth fighting for. I was determined to address these issues head-on, no matter the challenges that I faced.
*[... 9 more QA examples ...]*
**Question:** Abraham Lincoln, how do you define freedom?
**Answer:**

---

# F. Evaluation Details

**Generation Setup.** Unless otherwise specified (e.g., baseline sourcing), text generation uses nucleus sampling (top-$p$) with $p = 0.9$. For detoxification evaluations on RealToxicityPrompts (Gehman et al., 2020), we use the standard 10k test

prompts from Liu et al. (2021) and generate $k = 25$ continuations per prompt.

**Detoxification Metrics.** *Toxicity:* Assessed via Perspective API scores. We report average maximum toxicity over $k = 25$ generations, and the empirical probability of $\geq 1$ generation having toxicity $\geq 0.5$. *Perplexity(PPL):* Mean perplexity of generations using GPT2-XL. *Diversity:* Mean distinct $n$-grams ($n = 2, 3$), normalized by length, over $k = 25$ generations (Dist-2, Dist-3).

**GPT-4 Evaluation (Detoxification).** We used GPT4o-mini for supplementary evaluation. For each prompt, the 25 generated continuations were presented together. The model was asked to rate the set on toxicity, fluency, and diversity (1-5 scale) using the following prompt format:

---

**GPT-4 EVALUATION PROMPT & FORMAT**

I will provide you with a prompt, followed by 25 continuations of that prompt sampled from a model. Please provide an overall score for the set of continuations according to toxicity, fluency, and diversity, each on a scale of 1 to 5.
Output your evaluation in the following format:
**Evaluation for Toxicity:** (your rationale for the rating, as text)
**Toxicity rating:** (your rating, as a number between 1 and 5 where 1 is toxic and 5 is non-toxic)

**Evaluation for Fluency:** (your rationale for the rating, as text)
**Fluency rating:** (your rating, as a number between 1 and 5 where 1 is not fluent and 5 is fluent)

**Evaluation for Diversity:** (your rationale for the rating, as text)
**Diversity rating:** (your rating, as a number between 1 and 5 where 1 is not diverse and 5 is diverse)

---

**Role Quality Metric.** As ground truth for role quality is unavailable, we use the trained character-specific classifier itself as an evaluator. Role quality is measured as the average probability assigned by the target character's classifier to the generated texts.

**Political Relevance Metric.** Political content for the composition task (Section 5.5) is scored using the zero-shot classifier from Laurer et al. (2023); we report the mean score over generations.

