# OpenReview forum: "TRACE Back from the Future: A Probabilistic Reasoning Approach to Controllable Language Generation"
_ICML.cc/2025/Conference — ICML 2025 poster_

### Official Review · Reviewer_UD8f · 2025-03-12

**Overall Recommendation:** 3

**Summary:**

This paper introduces TRACE, a framework for controllable text generation that combines a distilled Hidden Markov Model (HMM) with lightweight classifiers to guide language model outputs toward desired attributes. The core idea is to compute the Expected Attribute Probability (EAP) tractably via HMM forward-backward inference, enabling efficient reweighting of next-token probabilities during decoding. TRACE aims to address the limitations of existing methods—such as computational expense, inflexibility, and reliance on sampling—by decoupling generation (via HMM) and control (via classifiers). The method is evaluated on detoxification, personalized language model adaptation, and compositional attribute generation. Empirical results demonstrate state-of-the-art detoxification performance, rapid adaptation to new attributes (e.g., 76 personalized roles trained in seconds), and seamless compositionality, with minimal decoding overhead (1.1× baseline latency).

**Claims And Evidence:**

The claims are generally supported by empirical evidence.

However, the claim that HMM quality directly impacts performance (Section 5.1) lacks rigorous analysis. While Table 1 shows TRACE (↓HMM) underperforms, the paper does not quantify HMM expressiveness (e.g., hidden state size, KL divergence between HMM and LM distributions). Additionally, the paper does not explicitly define "↓HMM."

**Essential References Not Discussed:**

NA

**Experimental Designs Or Analyses:**

Detoxification: Experiments on GPT-2 and Gemma-2B are thorough, but toxicity evaluation lacks diversity (e.g., no human judgments).

Personalization: Training classifiers on 300 samples per role (Section 5.2) may risk overfitting, and the results are only qualitatively validated (Table 2).

Compositionality: The independence assumption for combining attributes (Section 5.4) is untested for correlated and anti-correlated attributes.

**Methods And Evaluation Criteria:**

Methods: The HMM distillation + classifier framework is sensible for tractable EAP computation. However, the token-level factorized classifier (Equation 6) oversimplifies semantic attributes (e.g., style) and is not validated for non-factorizable tasks.

Evaluation: Human evaluation on toxicity and fluency may bring insights.

**Other Comments Or Suggestions:**

Suggestion: Include ablation studies on HMM Quality.

**Other Strengths And Weaknesses:**

While mostly well-structured, the decoding-time transformation (Section 5.1) is under-explained.

**Questions For Authors:**

Can TRACE handle non-factorizable attributes (e.g., coherence)? If not, what modifications are needed?

**Relation To Broader Scientific Literature:**

TRACE builds on:

- HMM-based control: Extends Ctrl-G from lexical to semantic attributes.

- Decoding-time control: Improves upon FUDGE and DExperts by replacing neural discriminators with tractable HMMs.

- Compositionality: Similar to energy-based methods (e.g., COLD) but avoids MCMC sampling.

**Theoretical Claims:**

The paper makes no formal theoretical claims. The HMM forward-backward algorithm and EAP derivation (Section 4.2) are standard but correctly applied.

---

> ### Author Rebuttal · Authors · 2025-04-01
>
> Thank you for the insightful feedback.
> ```
> Detoxification: Experiments on GPT-2 and Gemma-2B are thorough, but toxicity evaluation lacks diversity (e.g., no human judgments).
> ```
> We ran LLM-as-judge evaluations with GPT-4 to evaluate the toxicity, fluency, and diversity of the generated continuations on a scale of 1-5 (where 1 is toxic/nonfluent/nondiverse and 5 nontoxic/fluent/diverse), and took the average over the dataset. The results for the Gemma-2B based methods can be found in [Table](https://anonymous.4open.science/r/TRACERebuttal-97A4/gpt4_evaluations.pdf); these are consistent with the conclusions from the automated metrics.
> ```
> Personalization: Training classifiers on 300 samples per role (Section 5.2) may risk overfitting, and the results are only qualitatively validated (Table 2).
> ```
> We agree that training classifiers with only 300 samples per role may risk overfitting, but TRACE’s use of a factorized classifier may be less susceptible compared to neural classifiers. Following your suggestion, we added a quantitative evaluation of personalization performance against a prompting baseline, using the prompt “You are a role-playing assistant trained to embody characters with accuracy and authenticity. In this instance, you will assume the persona of {role_name}. Answer the question: {question}”. TRACE performs better in role quality as shown in [Fig](https://anonymous.4open.science/r/TRACERebuttal-97A4/role_quality.png).
> ```
> Compositionality: The independence assumption for combining attributes (Section 5.4) is untested for correlated and anti-correlated attributes.
> ```
> The independence assumption says that two attributes are independent conditional on the full text. This is fundamentally a modeling assumption regarding the uncertainty of the classifier given by the joint distribution over attributes given text. Intuitively, it says that “I believe this text is toxic with probability 0.7, and political with probability 0.4, and so I believe it is toxic and political with probability 0.28”. In order to test the hypothesis empirically, one would need a ground-truth probabilistic classifier that models the joint distribution over attributes given text, which does not typically exist.
>
> In particular, we do not require that two attributes are independent unconditionally (which would be violated with correlated or anti-correlated attributes). We demonstrate this empirically with the politics and nontoxicity attributes, where TRACE is able to enforce both politics and nontoxicity compositionally to the same level as each individually, despite the fact that these attributes appear to be anticorrelated.
> ```
> While mostly well-structured, the decoding-time transformation (Section 5.1) is under-explained.
> ```
> We will include a more detailed description to the revised version. We would be happy to answer any questions the reviewer may have about the decoding-time transformation.
> ```
> Can TRACE handle non-factorizable attributes (e.g., coherence)? If not, what modifications are needed?
> ```
>
> Please see our response to Reviewer 7o4g regarding the empirical performance on non-factorizable attributes. In summary, all attributes are to some extent non-factorizable, but TRACE obtains good performance even for (mildly) non-factorizable attributes.
>
> Though TRACE cannot currently handle heavily non-factorizable attributes except by approximation, we believe that the methodology of TRACE could be modified to directly capture non-factorized attributes. The key requirement is for the computation of EAP to be tractable with respect to a HMM. One extension could be to utilize a (weighted) sum of factorized classifiers, and then utilize linearity of expectation to compute EAP, though this incurs an increase in computational cost. More generally, the literature on tractable probabilistic models [1] describes more complex functions whose expectation can be computed efficiently with respect to a tractable distribution such as an HMM. Given the strong empirical performance of TRACE, we believe that investigating compute-efficient methods of relaxing the factorization assumption offers highly promising avenues for future work.
>
> [1] Khosravi et al. “On Tractable Computation of Expected Predictions” NeurIPS 2019
> ```
> Suggestion: Include ablation studies on HMM Quality
> ```
>
> We ran a study analysing the performance of TRACE at different points in the training/distillation process to evaluate the effect of HMM quality on toxicity reduction [Fig](https://anonymous.4open.science/r/TRACERebuttal-97A4/hmm_quality.pdf); the results show that the toxicity reduction improves with HMM log-likelihood.

---

### Official Review · Reviewer_UNDs · 2025-03-14

**Overall Recommendation:** 4

**Summary:**

The paper proposes a new method for controlled language modeling. Motivated by the compute and data inefficiency of previous solutions, the paper introduces a new method called TRACE, which uses conditional probabilities from an HMM to adjust token probabilities such that the text demonstrates desired attributes. The corresponding attribute classifier uses log-MSE on probabilities obtained through an oracle together with a probability transformation technique to make the distribution more likely to be bimodal. This technique can also be applied during decoding to improve results.
The method is evaluated on detoxification, character role play, and the generation of both political and non-toxic text.

**Claims And Evidence:**

Claim #1: the method reaches superior performance to the baselines. This is supported through table 1, although the training details of baselines are unclear.

Claim #2: the method can quickly adapt to new attributes. This is demonstrated on a role-playing task, where the LLM is adapted to speak like a specific character based on 200 texts of that character. This method is not compared to other baselines, so it is unclear how strong the results are.

Claim #3: the method is more compute efficient. Table 4 and 5 support this clearly.

Claim #4: TRACE can combine multiple attributes easily. This is somewhat supported by Table 3. However, again there is no comparison to a baseline so it is unclear how much of an achievement this is.

**Essential References Not Discussed:**

None

**Experimental Designs Or Analyses:**

The experimental design is probably correct, however, there are too little details given. See question section.

**Methods And Evaluation Criteria:**

The proposed method is reasonable and the evaluation benchmarks are solid. Obviously, there could always be more evaluations, e.g. on other types of attributes, but the amount of support in this paper is satisfactory.
As noted previously, the details of the experiments are insufficient.

**Other Comments Or Suggestions:**

* line 250: detofixication instead of detoxification
* Is log-MSE a standard method in this case? Can you cite a reference?
* The caption of Table 1 contains no explanation of why some fluency measures are striked-through. This is only in the text.
* Table 4: should probably say > 1 day for GeDi
* training time transformation seems so important it shouldn't be in Appendix A but rather be in the main body of the paper

**Other Strengths And Weaknesses:**

The paper lacks in clarity regarding the experimental results. See questions and comments below.

**Questions For Authors:**

* How are the baseline numbers obtained that are listed in Table 1? If you got them from the respective numbers, clarify so. If you trained them yourself, what did you do to ensure that the differences in performance are not due to differences in evaluation and/or details of the implementation that are not central to the method?
* How do the baselines perform on character role-playing and political and nontoxic text generation?
* How is character probability in Figure 3 measured? I cannot find any information in the text or the appendix.
* The impact of HMM quality is not well explained. What is the difference between TRACE and TRACE (HMM)? Don't all TRACE results use an HMM?
* Since this seems to be a crucial aspect of your method, what is the performance of the method without training-time transformation?

**Relation To Broader Scientific Literature:**

The paper proposes a method for controlled generation that is simple, efficient, and effective. It is related relatively well to the existing literature on this topic. Baseline evaluations are missing on two tasks, however.

**Theoretical Claims:**

I checked the derivation of the method and found no mistake.

---

> ### Author Rebuttal · Authors · 2025-04-01
>
> Thank you for the detailed feedback.
> ```
> Is log-MSE a standard method in this case? Can you cite a reference?
> ```
> The concept of log-MSE is somewhat analogous to mean squared logarithmic error in the context of regression. It shares the intuition of penalizing more heavily one type of misprediction than the other. In our case, if the classifier predicts 0.5, then under the cross-entropy loss we have the same penalty whether the true probability is 0.01 or 0.99. On the other hand, under log-MSE the penalty is $|ln(0.01) - ln(0.5)|^2 = 15.31$ versus $|ln(0.99) - ln(0.5)|^2 = 0.4666$, meaning the model is incentivized to conservatively skew towards nontoxicity. We are not aware of prior work specifically in the context of controllable generation / attribute classification which uses log-MSE.
> ```
> How are the baseline numbers obtained that are listed in Table 1? If you got them from the respective numbers, clarify so.
> ```
> The baseline results for PPLM, DAPT, GeDi, and DExperts were obtained from the DExperts paper; the results for FUDGE and MuCoLa were obtained from the MuCoLa paper; and the results for PPO and Quark were taken from the Quark paper. All of these follow the same experimental setup of DExperts. Regarding the DPO baseline that we trained and evaluated for GPT2-large and Gemma (see [Table](https://anonymous.4open.science/r/TRACERebuttal-97A4/detoxification.pdf)), we followed the evaluation setup of DExperts (that we also use for TRACE) and used the reference implementation for training, replacing their GPT2-medium model with Gemma-2B.
> ```
> How do the baselines perform on character role-playing and political and nontoxic text generation?
> ```
> The main purpose of these experiments is to provide evidence for the flexibility and efficiency of TRACE, with the empirical observations that (i) TRACE requires minimal training/decoding overhead for new attributes compared to baselines (Figure 3, Table 4, Table 5); and (ii) TRACE handles composition seamlessly (e.g. Detox, Pol and Detox+Pol rows of Table 3) with no additional overhead. In particular, the baselines would have difficulty adapting quickly to tens of new attributes or handling compositional attributes, requiring significant additional computational overhead for training and decoding (e.g. GeDi would require training class-conditional language models for combinations of attributes, and all of the role-playing characters, which takes 5 hours for each new attribute). This precisely demonstrates the unique position of TRACE within the literature.
>
> We added a comparison of TRACE and prompting the LLM with “You are a role-playing assistant trained to embody characters with accuracy and authenticity. In this instance, you will assume the persona of {role_name}. Answer the question: {question}”. (See [Fig](https://anonymous.4open.science/r/TRACERebuttal-97A4/role_quality.png)). TRACE outperforms prompting in role quality.
> ```
> How is character probability in Figure 3 measured?
> ```
> The character probability is measured by the linear classifier we trained on the characters; this is because we do not have a ground truth model/metric available.
> ```
> The impact of HMM quality: What is the difference between TRACE and TRACE (\downarrow HMM)? Don't all TRACE results use an HMM?
> ```
> The TRACE (\downarrow HMM) refers to the results of TRACE when using less data to train the HMM (500K vs 5M examples); the results show that a stronger HMM can provide stronger guidance with the same classifier to reduce toxicity further. Inspired by the reviewers’ comment, we ran a study analysing the performance of TRACE at different points in the training/distillation process to evaluate the effect of HMM quality on toxicity reduction ([Fig](https://anonymous.4open.science/r/TRACERebuttal-97A4/hmm_quality.pdf)); the results show that the toxicity reduction improves with HMM log-likelihood.
>
>
> ```
> Since this seems to be a crucial aspect of your method, what is the performance of the method without training-time transformation?
> ```
> TRACE with training time has a result of max tox 0.336, avg tox 0.2, fluency 33.87, diversity 0.86. As the reviewer notes, the transformation is crucial to produce a more bimodal distribution of scores, which also leads to significantly improved performance. This is necessary because the scores from the Perspective API do not align well with the inductive bias needed for controlled generation. Without this transformation, the linear classifier predominantly defaults to labeling text as nontoxic.

---

> > ### Comment · Reviewer_UNDs · 2025-04-07
> >
> > Thank you for your response. It made me somewhat more confident that this is a good paper, assuming that the responses will be reflected in the CR version of this paper. I raised my score to reflect that.

---

### Official Review · Reviewer_7o4g · 2025-03-14

**Overall Recommendation:** 3

**Summary:**

This paper proposes TRACE, a new algorithm for controllable generation that uses a hidden markov model and a small classifier to "look ahead" and reweight the language model's token probabilities. They compare to many other controllable generation methods and report promising results on two types of controllable generation: controlling for "toxic" text as well as "political" text.

**Claims And Evidence:**

Within domain, the benchmarking results are very compelling, as well as the training and inference time measurements. However, I find the advertisement of this method as a general improvement for controllable generation to be a large overclaim, since the method relies on token-level classifier predictions to determine whether text satisfies the control task. This method seems specifically designed to work for sentiment-like tasks such as politicality and toxicality, for which bag-of-words classifiers are sufficient. They did not test on other common controllability tasks, for example poetry generation, because those tasks require sequence-level reasoning.

**Essential References Not Discussed:**

The list of benchmarks in Table 1 is nearly comprehensive, but seems to be missing COLD (https://arxiv.org/abs/2202.11705), which is mentioned in the related work. Is there a reason why this comparison was omitted?

**Experimental Designs Or Analyses:**

The experiments seem correct, and in particular the inference-time analysis is useful, as well as the tradeoff measurement between fluency and control satisfaction. As noted multiple times, this work seems to be missing important benchmarks from the wider controllable generation literature.

**Methods And Evaluation Criteria:**

As noted several times, there are many other controllable text generation tests that are not shown here. A classic place to start would be with the three tasks from FUDGE (https://arxiv.org/abs/2104.05218): couplet completion, topic control, and formality.

**Other Comments Or Suggestions:**

- More details about the HMM would be useful
- Table 4 caption and Table 5 caption should explain more about the experiments – what datasets and context lengths were used, things like that.

**Other Strengths And Weaknesses:**

Strengths:
The method relies on distilling a GPT-like transformer LM to a small hidden markov model, which is interesting and novel.

Weaknesses:
- The mathematical formulation relies on a token-level classifier p(s | x_{1 ...n}) that's simply the sum of token-level predictions (Section 4.3). This seems like a huge problem as almost all control conditions of interest can not be modeled at the token level and require some level of contextual dependence. While interesting, this method is almost specifically designed for toxicity or sentiment-like tasks which are famously solvable with bag-of-words approaches.

**Questions For Authors:**

Is there a reason why you only evaluate on these specific domains (toxicity + political text)?

**Relation To Broader Scientific Literature:**

This paper contributes a new method to controllable text generation and compares to many other controllable generation methods. Notably absent is comparison across the same tasks.

**Theoretical Claims:**

I checked the mathematical derivations in Section 3 and Section 4, which seem correct.

---

> ### Author Rebuttal · Authors · 2025-04-01
>
> Thank you for the constructive feedback.
> ```
> Why evaluate on these specific domains (toxicity + political text)?
> ```
> In the related literature, papers have evaluated on many different tasks rather than (a set of) fixed common benchmarks. We selected toxicity evaluation as it is a widely recognized benchmark in the literature (e.g., DExperts). Additional experiments, including personalized LLM and political compositionality, were included to showcase TRACE’s efficiency and flexibility relative to computationally intensive methods.
>
> ```
> Additional controllable generation tasks (topic control, formality, couplet completion)
> ```
> Following your suggestion, we performed additional topic control experiments, comparing TRACE to the base model and GeDi (https://anonymous.4open.science/r/TRACERebuttal-97A4/topic_control.pdf). TRACE significantly improves over the baseline and outperforms GeDi on two of four topics, demonstrating effectiveness even with mild contextual dependencies. Formality is primarily considered a translation task rather than a generation task and thus was not evaluated. Couplet completion, which heavily relies on contextual dependencies, is discussed below.
> ```
> Limitations of token-level classifiers: …almost all control condition…require some level of contextual dependence. While interesting, this method is almost specifically designed for toxicity or sentiment-like tasks which are famously solvable with bag-of-words approaches.
> ```
> The reviewer is correct in saying that linear/factorized classifiers cannot perfectly capture all attributes. However, this is not a fundamental problem for our approach, for three main reasons:
> 1. Even when an attribute is nonlinear, exactly conditioning a linear classifier at generation/decoding time is effective - which is what we show in the toxicity and topic control experiments in which we are competitive with or beat existing approaches;
> 2. Our method has virtually no training or inference-time overhead, making it the only option when flexibility and latency are critical.
> 3. For tasks with heavy contextual dependence, one can still trade compute for accuracy.
>
> To validate point 1, we ran experiments investigating the difference in classification performance between our factorized classifiers and neural classifiers for these tasks (https://anonymous.4open.science/r/TRACERebuttal-97A4/nonfactorizable.pdf). On all tasks (including toxicity and politics), there is a gap in classification performance (as measured by cross-entropy with oracle scores), illustrating that *none of these attributes are completely factorizable*. Despite this, TRACE achieves superior performance on these mildly non-factorizable tasks. The reason is that conditional generation is a computationally hard problem, even for a factorized attribute. TRACE addresses this computational hardness of exact conditional generation by employing a lightweight, tractable approximation (HMM + linear classifier), and in doing so can outperform approaches that rely on neural classifiers but cannot effectively approximate expected attribute probability (EAP).
>
> For point 3, for tasks/attributes with heavy contextual dependence, e.g. couplet completion, we can employ TRACE to condition on the “linear part” of the attribute, and filter the generated outputs post-hoc using a more powerful classifier. Alternatively, we believe there is scope to relax the factorized classifier assumption. The key requirement is for the computation of EAP to be tractable with respect to a HMM. One extension could be to use a (weighted) sum of factorized classifiers, and then use linearity of expectation to compute EAP, though this incurs an increase in computational cost. More generally, the literature on tractable probabilistic models [1] describes more complex functions whose expectation can be computed efficiently with respect to a tractable distribution such as an HMM. Given the strong empirical performance of TRACE, we believe that investigating compute-efficient methods of relaxing the factorization assumption offers highly promising avenues for future work.
>
> ```
> COLD appears in Related Work, but not detoxification results table?
> ```
> We chose to report results from papers testing on the toxicity reduction task for RealToxicityPrompts following the widely-used evaluation setup of DExperts; however COLD does not have any results on this task.
> ```
> More details about the HMM
> ```
> Thanks, we will add these to the paper. The HMM used in the experiments has 4096 hidden states with around 223M parameters. We will add details of the distillation procedure to the Appendix.
> ```
> More experiment details on Table 4 and Table 5 captions – what datasets and context lengths were used.
> ```
> We will add these details to the captions. The statistics for TRACE were gathered through the toxicity (RealToxicityPrompts) and personalization experiments.
>
> [1] Khosravi et al. “On Tractable Computation of Expected Predictions” NeurIPS 2019

---

### Official Review · Reviewer_Qbev · 2025-03-16

**Overall Recommendation:** 3

**Summary:**

Large language models (LLMs) are increasingly being deployed in real-world applications, and the need to control their outputs to align with human values is becoming more important. However, current autoregressive models struggle when attempting to control their generations. This paper proposes a technique called TRACE (Tractable Probabilistic Reasoning for Adaptable Controllable Generation) that can manage LLMs' outputs using a Hidden Markov Model (HMM). The authors' framework is tractable and lightweight compared to other proposed techniques in the literature. They evaluate their proposed method on three tasks: detoxification, personalized LLMs, and compositional attributes, and demonstrate that their approach outperforms baseline algorithms in the literature.

**Claims And Evidence:**

The paper claims that RLHF is expensive and risks degrading the fluency or diversity of generated text. The authors conducted experiments to support their claims, but Table 1 contains several numbers struck out with asterisks, and it's unclear what that signifies. Additionally, the results for Gemma-2B with other algorithms are missing, which means there is insufficient evidence to support this claim. In particular, GPT-2 is known to be an inferior model compared to Gemma, Llama, and more recent LLMs, so experimenting with all algorithms under a modern LLM would provide better evidence. If the baseline algorithm numbers were taken from previous papers, then these results might be misleading, especially since the infrastructure supporting algorithms has changed tremendously.

**Essential References Not Discussed:**

- Is reinforcement learning (not) for natural language processing?: Benchmarks, baselines, and building blocks for natural language policy optimization by Ramamurthy et al. 2023
- Plug & Play Generative Networks: Conditional Iterative Generation of Images in Latent Space by Nguyen et al 2016
- Back to the Future: Unsupervised Backprop-based Decoding for Counterfactual and Abductive Commonsense Reasoning by Qin et al 2020
- Technical Report: Auxiliary Tuning and its Application to Conditional Text Generation by Zeldes et al 2020

**Experimental Designs Or Analyses:**

The authors' experimental design is generally sound; however, I am currently unclear about the meaning of the lines through the numbers in Table 1. Additionally, I am uncertain whether the authors conducted all the experiments themselves or if the numbers were sourced from previous papers. The results also lack qualitative measures, such as human or AI model evaluations. The metrics presented capture only one aspect of the problem. Using perplexity to assess fluency is insufficient for evaluating the quality of the generated text. Furthermore, it is not clear how toxicity was scored; if a external model provides the scores their is not prompt templated provided in the appendix.

**Methods And Evaluation Criteria:**

The paper evaluated their proposed algorithm on three datasets: detoxification, personalized LLMs, and compositional attributes. Evaluating on three popular controllable generation tasks addresses the problem the paper is trying to tackle.

**Other Comments Or Suggestions:**

None

**Other Strengths And Weaknesses:**

No

**Questions For Authors:**

a) How does the proposed approach compare to Zeldes et al., 2020?

b) Why is equation 2 generally intractable? If s represents text, then equation 2 essentially corresponds to the first term of equation 3, which is simply the language model's next token distribution.

c) How does the most naive baseline of simply prompting the model perform? Most personalization papers have this baseline incorporated in their results. (see Jang et al. 2023)

d) The text below Equation 5 says, conditional independence no longer holds when a generic attribute s is introduced, as it depends on all of x . However, in Equation 6, a conditional independence assumption is made in order to factorize over all of x. I am confused about why this factorization is considered a fair assumption when conditional independence was not.

e) Equation on line 176, p(s|x_{1:n}) does not makes sense because x_{1:n} is not past into function.

f) Did you train the baseline algorithms in the GPT2 row yourself? If not, what papers did you get the numbers from?

g) How does PPO, Quak, and DPO perform with Gemma-2B?

h) How does the model perform when conducting GPT-4 evaluations for text quality and toxicity qualitative analysis? Currently, you are using perplexity as a metric, but that measure is known to be quite misleading. Similarly, output perplexity has its own set of drawbacks as well.

- Technical Report: Auxiliary Tuning and its Application to Conditional Text Generation by Zeldes et al. 2020

- Personalized Soups: Personalized Large Language Model Alignment Via Post-Hoc Parameter Merging by Jang et al. 2023

**Relation To Broader Scientific Literature:**

The authors are addressing a challenging problem in the scientific literature. In particular, the ability to control generative models has been a difficult issue that researchers have been trying to solve for quite some time. The authors observe that, unlike current solutions, HMM can provide a more tractable and lightweight solution with strong practical performance.

**Theoretical Claims:**

N/A

---

> ### Author Rebuttal · Authors · 2025-04-01
>
> Thank you for the detailed review.
> ```
> Table 1 contains several numbers struck out with asterisks, and it's unclear what that signifies.
> ```
> We explained this in Lines 261-274 (left) in the main text but forgot to add it to the Table caption - we will do this in the revision.
>
> ```
> a) Comparison to Zeldes et al. (2020).
> ```
> Zeldes et al. (2020) combines the logits of a base LM $p(x_t | x_{< t})$ with the logits of an auxiliary LM that models the conditional next token distribution $p(x_t|x_{<t}, \alpha)$. The distinguishing feature of TRACE (also compared to approaches such as GeDi) is that, instead of training an auxiliary neural model for each attribute, we simply use one HMM as an approximation to the base LM, and condition it on the linear classifier of any attribute to guide the base LM’s generation.
> ```
> b) Why is equation 2 intractable? If s represents text, equation 2 corresponds to the first term of equation 3, which is simply the language model's next token distribution.
> ```
> In Equation 2, $s$ represents an attribute, rather than a text prefix. This attribute is a function of the entire text $x_{1:n}$, and as illustrated in the following equation computing this requires an exponential sum over future continuations.
> ```
> c) Naive prompting baseline?
> ```
> We added a comparison of TRACE and prompting the LLM with “You are a role-playing assistant trained to embody characters with accuracy and authenticity. In this instance, you will assume the persona of {role_name}. Answer the question: {question}”. https://anonymous.4open.science/r/TRACERebuttal-97A4/role_quality.png
> ```
> d) The text below Equation 5 says, conditional independence no longer holds… I am confused about why this factorization is considered a fair assumption when conditional independence was not.
> ```
> We are not saying that conditional independence is an unreasonable assumption, but rather deriving a condition (factorized classifier) under which it holds, and noting that it does not hold in general.
> ```
> e) Equation on line 176, p(s|x_{1:n}) does not makes sense because x_{1:n} is not past into function.
> ```
> By $x_{1:n}$ we mean the entire text, which is the concatenation of the prefix $x_{<t}$ and current token $x_t$ (which are passed in), and the future continuation $x_{>t}$ (which appears in the summation). We will revise to make this clearer.
> ```
> f) Source of GPT2 baseline numbers.
> ```
> The baseline results come from:
> - PPLM, DAPT, GeDi, DExperts: from DExperts paper.
> - FUDGE, MuCoLa: from MuCoLa paper.
> - PPO, Quark: from Quark paper.
>
> All of these follow the same experimental setup of DExperts. For fair comparison, we fine-tuned GPT2-large using DPO (official implementation of Lee et al. [1]) with the same DExperts setup, resulting in avg max toxicity 0.180, toxicity prob. 0.03, fluency 21.59, dist-2 diversity 0.76, and dist-3 diversity 0.78.
>
> ```
> g) How does PPO, Quak, and DPO perform with Gemma-2B?
> ```
> We trained DPO on Gemma-2B using [1]'s official code; results are at https://anonymous.4open.science/r/TRACERebuttal-97A4/detoxification.pdf. Consistent with our GPT-2 experiments, DPO improves fluency over the base model while diversity is significantly reduced. Meanwhile, TRACE achieves superior toxicity reduction while largely maintaining Gemma's original fluency and diversity. Unfortunately, we had trouble adapting PPO and Quark to Gemma during the rebuttal period due to GPT-2 specific codebases.
> ```
> h) GPT-4 evaluation for text quality and toxicity.
> ```
> We utilize perplexity to evaluate the fluency of text following the common practice among the baselines. While perplexity may not align perfectly with LLM-as-a-judge or human evaluations of quality, it carries important and distinct scientific value in measuring the deviation from the text distribution of the base model. For instance, the non-RL baselines use the same decoding strategies (top-$p$ sampling), enabling direct comparison of the different approaches. Meanwhile, the RL methods modify the base model which significantly impacts the distribution as becomes apparent from the fluency (perplexity) and diversity metrics, but may not be apparent from an LLM-as-a-judge or human evaluation. For toxicity evaluation, we use the Perspective API following existing practice in previous works.
>
> We also conducted evaluations with GPT-4 as LLM-as-a-judge to evaluate the toxicity, fluency, and diversity of the generated continuations on a scale of 1-5 (where 1 is toxic/nonfluent/nondiverse and 5 nontoxic/fluent/diverse), averaged over the dataset. The results (https://anonymous.4open.science/r/TRACERebuttal-97A4/gpt4_evaluations.pdf) are consistent with the evaluations given by the automated metrics. In particular, both TRACE and DPO effectively reduce toxicity, but TRACE maintains similar fluency and diversity of generations to Gemma, while DPO distorts the distribution.
>
> [1] Lee et al. “A Mechanistic Understanding of Alignment Algorithms: A Case Study on DPO and Toxicity” ICML 2024

---

### Decision · Program_Chairs · 2025-05-01

**Decision:**

Accept (poster)

**Comment:**

This paper introduces TRACE, a framework for controllable text generation that combines a distilled HMM with classifiers to guide language model's generations. TRACE aims to address the limitations of existing methods—such as computational expense, inflexibility, and reliance on sampling—by decoupling generation (via HMM) and control (via classifiers). Empirical results demonstrate good performance and rapid adaptation to new attributes.

After the rebutall, reviewers are in general all positive about the paper. Before rebuttal there were concners around comparisons to more baselines, evaluating using GPT4 based metrics, additional experiments on other base models, the impact of the HMMs, and how to handle non-factorizable attributes. In the rebuttal the authors responded with additional expeirments and reviewers acknowledged that their concerns are adressed.